# A Web-Based Automated Image Processing Research Platform for Cochlear Implantation-Related Studies

**DOI:** 10.3390/jcm11226640

**Published:** 2022-11-09

**Authors:** Jan Margeta, Raabid Hussain, Paula López Diez, Anika Morgenstern, Thomas Demarcy, Zihao Wang, Dan Gnansia, Octavio Martinez Manzanera, Clair Vandersteen, Hervé Delingette, Andreas Buechner, Thomas Lenarz, François Patou, Nicolas Guevara

**Affiliations:** 1Research and Development, KardioMe, 01851 Nova Dubnica, Slovakia; 2Research and Technology Group, Oticon Medical, 2765 Smørum, Denmark; 3Department for Applied Mathematics and Computer Science, Technical University of Denmark, 2800 Kongens Lyngby, Denmark; 4Department of Otolaryngology, Medical University of Hannover, 30625 Hannover, Germany; 5Epione Team, Inria, Université Côte d’Azur, 06902 Sophia Antipolis, France; 6Institut Universitaire de la Face et du Cou, Centre Hospitalier Universitaire de Nice, Université Côte d’Azur, 06100 Nice, France

**Keywords:** cochlea, cochlear implant, image analysis, computed tomography, machine learning, deep learning, image segmentation, 3D model, tonotopic mapping, visualization

## Abstract

The robust delineation of the cochlea and its inner structures combined with the detection of the electrode of a cochlear implant within these structures is essential for envisaging a safer, more individualized, routine image-guided cochlear implant therapy. We present Nautilus—a web-based research platform for automated pre- and post-implantation cochlear analysis. Nautilus delineates cochlear structures from pre-operative clinical CT images by combining deep learning and Bayesian inference approaches. It enables the extraction of electrode locations from a post-operative CT image using convolutional neural networks and geometrical inference. By fusing pre- and post-operative images, Nautilus is able to provide a set of personalized pre- and post-operative metrics that can serve the exploration of clinically relevant questions in cochlear implantation therapy. In addition, Nautilus embeds a self-assessment module providing a confidence rating on the outputs of its pipeline. We present a detailed accuracy and robustness analyses of the tool on a carefully designed dataset. The results of these analyses provide legitimate grounds for envisaging the implementation of image-guided cochlear implant practices into routine clinical workflows.

## 1. Introduction

Cochlear Implants (CI) are, to this day, the most successful neural interfaces ever engineered judging by their functional outcomes benefits, gains in quality of life, or widespread adoption in standard clinical practice [1]. More than 700,000 CI users worldwide have been eligible for and are undergoing CI therapy because of severe or profound deafness [2]. CI systems are neuroprosthetic devices generally composed of two parts. The first part is an external device called the sound processor and is usually worn behind the ear. It is responsible for real-time sensing, processing, and transmitting acoustic information (i.e., sound) to the other, internal, surgically implanted part of the system. This second part is in charge for transmitting the encoded acoustic information content to the auditory nerve by way of trains of electrical impulses delivered through an electrode array placed in the cochlea [2]. CI systems therefore bypass the cochlea altogether and replace the natural hearing mechanism with what is often referred to as “electrical hearing”.

Despite its large overall success, CI therapy still presents significant shortcomings. In particular, documented clinical outcomes remain variable and generally not fully predictable. Additionally, perceptual adaptation to CI hearing, even when functionally successful in terms of speech recognition and communication abilities, often remain unsatisfactory when it comes to real-life scenarios, including complex, spatial, and musical soundscapes [1]. A large body of knowledge points to anatomical factors and our current limited ability to assess patient-specific cochlear anatomy (pre-implantation) and its relation to CI electrode placement (post-implantation) as impediments to the development of more adapted best practices in surgical and audiological CI therapy. The intrinsic inter-individual variability of inner ear anatomy, for instance, compounds the challenge to predict the insertion dynamics of a specific CI electrode, making it difficult to plan and predict how deep a surgeon may expect to insert the CI electrode, which may have consequences on the low-frequency percepts that the implant may be able to elicit—also known as a consequence for the preservation of residual hearing. Likewise, the challenge of assessing where exactly the electrode contacts lay within the cochlea post-operatively prevents a CI device fitting/programming that takes into account the natural tonotopicity of the spiral ganglions lining up the cochlea or the consideration of the fitting parameters set for the contra-lateral ear in bilateral CI users [3,4,5]. A common denominator to these aspects is, therefore, the need for an intimate assessment of individual anatomy and geometry of cochlear structures and CI electrode placement relative to these structures in individuals from various clinical population eligible for CI therapy. Importantly, if some of the mechanisms at play in limiting CI therapy performance outcomes (whichever ones we look at) are known, much obscurity remains as to how to harness individual anatomical information to optimize and personalize CI therapy in relevant clinical populations.

Nautilus is a web-based research-grade tool that allows the automated, accurate, robust, and uncertainty-transparent delineation of the cochlea, scala tympani (ST), scala vestibuli (SV), and of the electrode arrays with tonotopic mapping from conventional computed tomography (CT) and cone-beam computed tomography (CBCT) images (see Figure 1).

### Background

The development of an automated imaging pipeline enabling the exploration of cochlear anatomy in clinical populations represents a significant challenge. The cochlear structures relevant to CI therapy, specifically the ST and SV, and the CI electrode array cannot always be easily delineated from clinical CT or CBCT images due to low image contrast and poor resolution. This prevents the manual delineation of ST and SV, which would anyway be a time-consuming, error-prone, and inconsistent process. More reasonably, semi- and fully automatic frameworks have been proposed to segment the cochlear bony labyrinth from pre-operative CT images. Earlier works focused on traditional segmentation techniques, such as level-set and interactive contour algorithms [6,7]. However, these required user input, were computationally time-consuming, and often led to incomplete segmentations. Recent works have focused on designing fully automatic convolutional neural networks capable of handling the intricate anatomy of the bony labyrinth [8,9,10,11]. The bony labyrinth is generally well identifiable in clinical CT or CBCT images, but its robust segmentation remains a challenge if one is to process images acquired with different scanners and image acquisition parameters, which may manifest in ranges of image resolution, contrast, and noise. Provided with a delineation of the bony labyrinth, various techniques permit the estimation of important metrics relevant to CI implantation, such as the cochlear duct length (CDL), which serves as an indicator of general cochlear size and what depth of insertion is reasonable to try to reach for that specific cochlea. The CDL and other metrics also enable the computation of normalized tonotopic frequencies according to Greenwood [12], Stakhovskaya [13], or Helpard et al. [14].

For all the information that can be gained from a segmentation of the bony labyrinth, many clinical questions call for the differentiation of ST from SV within the labyrinth. In this case, the automated image processing task becomes much more complex, since ST and SV are generally not visible in clinical CTs or CBCTs. Consequently, various atlases or shape models derived from temporal bone micro-CTs (µCTs) have been proposed to infer a ST/SV differentiation within the bony labyrinth when exploiting a clinical image [15,16,17,18,19,20]. The delineation of ST and SV is interesting in that CI implantation is preferentially done within ST as implantations or translocations in SV have been associated with observations of auditory pitch reversals and poorer speech intelligibility [21,22].

Post-operatively, CT imaging can provide information about the positioning of each electrode contact within or in the vicinity of the cochlea. However, the exploitation of post-operative CT/CBCT images is often compromised by metal artifacts emanating from the electrodes but generally affecting the region of interest around the electrodes enough so as to prevent the delineation of the bony labyrinth. Therefore, the post-implantation reconstruction of the CI electrode within cochlear structures often requires harnessing both the pre-operative and post-operative scans. Vanderbilt University’s group first proposed to independently segment intra-cochlear structures from pre-operative images using active shape models, followed by detection of the electrode array midline from post-operative imaging before combining pre- and post-operative information through a rigid registration [23]. They also proposed to take advantage of the left/right symmetry of inner-ear anatomy by utilizing the pre-operative image of the normal contra-lateral ear for cochlear structure delineation for cases where pre-operative CT images were not available [24]. Granting the successful reconstruction of electrode placement within cochlear structures, the characteristic frequency (CF) at each contact can legitimately be computed at the estimated corresponding place on the organ of Corti (OC) [12] or at the nearest spiral ganglion (SG) [13,14,25]. The accurate inference of the relative position between an electrode and the basilar membrane (BM) lining up the ST can also enable the assessment of the potential translocation of the electrode in SV or inferential predictions of the degree of traumaticity of the insertion, e.g., if the electrode were to have either elevated or ripped through the BM and entered the SV. Although state-of-the-art research on cochlear imaging has resulted in imaging pipelines that do display accuracy levels that can warrant their use in specific settings, these pipelines have generally not been subject to a strict robustness evaluation: their ability to deal with images of heterogeneous quality as one may expect to have to deal with when working on datasets obtained across different clinical centers. Searching to facilitate the exploration of clinical questions related to the anatomical and geometrical considerations of CI therapy, Nautilus enables the automated, accurate, robust, and transparent-on-uncertainty segmentation of the cochlear bony labyrinth, ST, and SV from pre-operative CT/CBCTs. Post-operatively, Nautilus enables the automated identification and reconstruction of the electrode arrays within the cochlear structures extracted from the pre-operative image. This tool computes a range of metrics relevant to both surgical and audiological research in CI, including the characteristic frequencies at each electrode contact. Nautilus’ predictions have been evaluated against several datasets annotated by experts and demonstrate state-of-the-art accuracy. Importantly, Nautilus was designed and stress-tested against images spanning a range of resolution, contrast, and noise, which results in its robust applicability, especially for a set of image input specifications that promote success, as we discuss later. Finally, the tool intends to transparently notify users of possible processing failures or complications using a set of caution flags to allow for the rejection of data points that may otherwise bias analysis.

## 2. Methods

Nautilus aims to be a gateway to advanced cochlear analysis. To maximize its availability, it has therefore been designed as a web application accessible via any modern web browser (e.g., Mozilla Firefox, Google Chrome, or Microsoft Edge) with no need for additional installation nor excessive requirements on the hardware. The data processing happens transparently on a cloud computing service. An overview of the processing pipeline can be seen in Figure 2, with Figure 3 illustrating the intermediary outputs of the process.

### 2.1. Data Upload and Pseudonymization via a Web-Based Frontend

Each user can create their private collection of images and associate each image to a specific case/individual. For each case, a unique anonymous identifier is generated upon creation. Once the image (most of the standard medical imaging formats are admissible (e.g., DICOM, NIFTI, MHA), as they can be loaded by ITK [26]) is loaded on the local browser, the image metadata (if any) are cleared of all personal identifiable information (PII). The user must then inform the laterality of the cochlea (left or right), whether it is a pre- or post-operative scan, and roughly place a cross on the targeted cochlea so as to allow the cropping and upload of a region of interest (ROI) from the original (albeit anonymized) image. After the data are uploaded, a processing job is queued and handled by the backend as soon as required computing resources become available.

### 2.2. Cochlear Landmarks and Canonical Pose Estimation

Cochlear pose estimation is essential to determine an initial orientation of the cochlea within the image and serves for image visualization in the standardized views [27]. The estimation of cochlear pose is also used for inferring the characteristic equation of the modiolar axis of the cochlea, which, in turn, is used to derive a number of metrics. We estimate the cochlear pose from a set of three automatically estimated landmarks—the center of the basal turn of the cochlea (C), the round window (RW—defined at its center), and the apex (Ap—defined at the helicotrema), as prescribed in [16]. Ap and C form the modiolar axis, which coincides with the z-axis. The basal plane passes through the RW, which defines the direction of the x-axis. The origin of the canonical reference coordinated is the intersection of the basal plane and the modiolar axis. Finally, the remaining axis is chosen such that the angle increases as we follow the cochlear duct starting from 0deg at the RW. The canonical reference frame allows Nautilus’ users to consistently compare cochleae of different sizes and allows equal treatment for both left and right cochleae.

A number of approaches have been proposed to estimate the landmarks or the pose, including registration and one-shot learning [28] or using regression forests to vote for the location of the landmarks [29]. More recently, reinforcement learning methods [30,31,32] have also been used to efficiently locate landmarks or to generate clinically meaningful image views [33] and, relevantly for our domain of application, to locate cochlear nerve landmarks [34]. Heatmap-based approaches consistently demonstrate robustness, explainability, and computational efficiency and offer an elegant form of uncertainty modelling and failure detection [35]. They do, however, sometimes have difficulties locating landmarks present around the image borders. We employ a conventional U-Net convolutional neural network architecture [36] as implemented in [37] with three output channels, one for each landmark. We modeled each landmark with a Gaussian heatmap and trained the network to map the input image to the three target heatmaps simultaneously. Our network architecture (detailed in the Appendix A) has 3 encoding blocks, 8 channels after the first layer and 16 output channels for the final feature map before the final projection onto the 3 heatmap channels (see Appendix A).

Our training set consists of an assortment of 279 pre- and post-operative clinical CT and CBCT images obtained from diverse sources. Our landmark detection block must be capable of handling (and was therefore trained on) both pre- and post-operative images. It is, however, significantly more difficult to accurately annotate C, RW, and Ap on the post-operative images due to the metallic artifacts. As a workaround, the pre-operative images were registered with the post-operative images, and the landmarks from pre-operative images were transported onto the post-operative images.

For training and inference, we resampled the input images to isotropic 0.3 mm spacing and normalized the intensities between the 5–95% percentile to 0–1 with no clipping. To increase the variability of our training set, we randomly sampled from a combination of data augmentations, such as random noise, flipping in all three dimensions, Gaussian blurring, random anisotropy [38], rigid transformations, and small elastic deformations as implemented by the TorchIO library [39]. Similarly to [40], we have observed that focal loss worked particularly well for sufficiently accurate landmark detection. During the inference, we transformed the predictions with the sigmoid activation to normalize them between 0 and 1, and for each output, we pick the mode of the output distribution (the hottest voxel of the heatmap) as the corresponding landmark.

### 2.3. Segmentation of Cochlear Structures

Nautilus is built with cochlear surgery planning, evaluation, and audiological fitting in mind. Therefore, in the current version, we focus on segmenting the two main cochlear ducts—ST and SV—and compute relevant measurements from these structures as others before us [41]. At a later stage, the delineation of ST and SV serves to relate the placement electrode array placement within the cochlea and infer information such as the characteristic frequency of each electrode contact [23]. An accurate and robust segmentation of ST and SV is therefore critical. Recent approaches based on convolutional neural networks have shown the most promise. Nikan et al., for instance [9], segmented various temporal bone structures including the labyrinth, ossicles, and facial nerve. Most of the cochlear segmentation approaches perform remarkably well on the cochlea and neighboring structures. They do not, however, separate the scalae [8,42], nor do they estimate the position of the BM, the delicate structure responsible for the transduction of mechanical waves within the cochlea into trains of electrical impulses, an essential structure to preserve in anticipation of restorative therapeutic advances. The separation of the scalae on clinical CTs is challenging as ST and SV are not discernible on clinical scans, mainly due to limited image resolution and contrast. To circumvent this issue, a shape model is often used to serve as a priori information on ST/SV distinction within the cochlear labyrinth. Recently, atlases [43] and a hybrid active shape model combined with deep learning [44] have been used with success for the separation of the scalae.

We used a pre-operative image of the implanted cochlea as the reference image for segmentation. Nautilus uses an approach similar to [44], which merges deep learning for appearance modelling with a strong shape prior constraining the final segmentation [45]. Instead of an active shape model, we build on top of a well-validated Bayesian joint appearance and shape inference model [20,46]. The parameters of this shape model were tuned and validated on µCT data. The model can then serve as a strong prior constraining the final output for the lower-resolution clinical CT images. This approach provides a probabilistic separation of ST and SV even in images of poor resolution. We provide an estimate of the BM location from the intersection of ST and SV’s probability maps. Demarcy et al.’s original Bayesian framework proposed to model the foreground and background appearance (i.e., intensity) as mixtures of Student distributions. We observed that this initialization is fairly sensitive to the type of scanner used for image acquisition and to image quality despite using normative Hounsfield units. To achieve better generalization, we therefore replaced the original appearance model with a trained convolutional neural network [36].

Similarly to our landmark detection approach, we used a reference 3D U-Net implementation of MONAI [37] with 6 encoding blocks, 8 output channels after the first layer (see Appendix A), and PReLU as the activation function and trained it on 130 images. We normalized the data by resampling the images to 0.125 mm spacing and rescaled the intensities such that the 5th and 95th percentile of the intensity distribution of each image were mapped to 0 and 1. In addition to augmentations used for landmark detection, we used random patch swapping [47] to increase the robustness to artifacts and force the network to learn a stronger shape prior. The model was trained on 128 × 128 × 128 patches with the AdamW [48] optimizer minimizing the Dice focal loss [37,49].

A large number of the metrics we extract from both pre- and post-operative processes depend on reliable estimation of the cochlear ducts’ centerline. Because our segmentation of ST and SV is based on a parametric shape model [46], extracting an approximate centerline is straightforward. We then refine this curve and estimate ST and SV centerlines from cross-sections of the segmentations along this curve. At each cross-section, we estimate the coordinates of the lateral wall landmark as the furthest point on the ST from the modiolar axis, OC at 80% of the distance to the LW [13], and the SG offset by −0.35 mm both radially and longitudinally from the modiolar wall landmarks (i.e., the point on the ST closest to the modiolus) as an approximation of Rosenthal’s canal.

### 2.4. Electrode Depth-to-Angular Coverage Prediction

The centerline can be discretized based on angles (in cylindrical coordinates), which can be used to predict a priori the angular coverage an electrode array is expected to reach as a function of the number of electrodes inserted beyond the RW. Shurzig et al. [50,51] proposed an ideal trajectory for the electrode, to be computed by subtracting the radius of the electrode from the radius of the cochlear spiral. A retrospective analysis of our predictions carried out on 98 images from our clinical dataset hinted that, on average, the CI electrode only follows an ideal trajectory after hitting the lateral wall around 150 deg. This observation leads us to propose the following statistical predictive model:(1)$$\delta_{i}=\rho -\left\{\begin{array}{cc} 1.3-0.007\theta_{i}, & \mathrm{if}\;i\leq 150^{\circ }\\ r_{i}, & \mathrm{otherwise} \end{array}\right.$$
where ρ is the radius of the centerline in cylindrical coordinates, and *r* and θ represent the radius of the *i*th electrode. Figure 4 depicts the angular errors based on Equation (Equation 1). Our predictions fall, on average, within $20^{\circ }$ of the observed insertion angular coverage (n = 58).

### 2.5. Registration of the Pre- and Post-Operative Images

To evaluate the electrode array placement within the cochlea, we need to be able to fuse the segmentation of the pre-operative scan and the electrode contacts of the post-operative scan to the same reference coordinate system. Although the post-operative scan is deteriorated by the metallic artifacts generated by the electrode contact, it still represents the bony structures somewhat similarly to what is seen in the pre-operative image. Rigid transformation is therefore possible for aligning pre- and post-operative images. We first pre-align the pre- and post-op image pair into their canonical poses with the previously estimated landmarks and fine-tune the final transform using the Elastix package [52,53]. We have observed that even for CT or CBCT images in Hounsfield units, the Advanced Mattes mutual information [54] with 64 histogram bins performs adequately. Invalid voxels (usually found at the boundaries of the image) and metallic artifacts in all voxels with HU>2500 are masked out and not used for computing the similarity.

### 2.6. Electrode Array Detection

The electrode array detection starts with the estimation of the 3D coordinates for each of the 20 electrode contacts before the subsequent evaluation of their placement, e.g., with relative distances from relevant cochlear structures such as SG, MW, LW, BM (distances which could presumably be used to infer an indicator of traumaticity [55]). The reconstruction of the electrode array can also help with the visual inspection and assessment of complications such as kinking, tip fold-over, or buckling [56]. Most of these patterns are difficult to identify on 2D images [57], and the 3D processing approaches provide significant advantages. Various approaches can be used to locate electrode contacts. Measuring peaks of an image intensity profile is a straightforward method [58]. When these peaks are less discriminative, modelling intensity and shape with Markov random fields can help [59], and so can morphological or filtering approaches with handcrafted rules [23,60,61] or graph-based approaches [62]. Many of these approaches work well when the image resolution is fine and the contacts are well resolved, with sufficient contrast and limited metallic artifacts, no significant kinking or tip fold-over; they often can be well tuned to a particular set of scanners. With our heterogeneous dataset, the evaluated methods suffer under uneven image quality and artifacts of various appearances. We used machine learning to enhance and detect the electrode contacts of the array and to generalize over differences in appearance and image quality between the different imaging vendors. We have designed a pipeline similar to our landmark estimation similar to [63] and trained a U-Net [36,37] to estimate the likelihood of a voxel being a center of a contact. However, in addition to the contact probability estimation, our network performs two additional tasks, which share a common feature extraction backbone (see Appendix A). For training, we annotated a dataset of 106 post-operative images with ITK-SNAP [64] containing all the individual electrodes (1–20) and lead wires (where visible). From the annotations, we generated 3 different target labels: electrode location heatmap common for all electrodes (with value 1 at the centers of the electrodes and 0 away from them). By connecting electrode coordinates, we constructed a curve, which we turned into a probability map for the electrode array, and lastly, we created a discrete label map with 5 classes (background, proximal electrode, mid-electrode, distal-electrode, and lead wire) used for semantic segmentation of the post-op images.

During the inference, we first estimated the contact probabilities and considered all peaks to be contact candidates. To create an electrode array out of this unsorted set of candidates, we started with the two most central points. We then iteratively fit a cubic B-spline to the already existing set and extrapolated at the two ends to search for the next probable point until no further expansion was plausible. This gave us a sorted array of contacts. To determine the final order, we assumed that the electrode array enters the cochlear around the round window and ascends along the cochlear duct towards the apex, i.e., the signed distance to the basal plane of the first contact should be smaller than that of the last most distal contact. We have observed that this strategy performs well even in the presence of mild to moderate aforementioned electrode array insertion complications. The lead wire is then estimated from the semantic segmentation by fitting a curve to the skeleton of the closest wire-like object near the first contact. This can serve to provide a more reliable estimation of the insertion angle [65].

This electrode array detection block is designed to operate on clinical CT and CBCT images, with, for the best performance, images of resolution of 0.3mm or finer with little anisotropy. The electrode array detector has currently been tuned for and tested with the CLA and EVO electrode array from Oticon Medical (24 mm long with 20 electrodes with 1.2 mm pitch and diameter ranging from 0.5 mm proximally to 0.4 mm distally) [66]. There is, however, no significant limitation to using it for models from other vendors (see Figure 5).

### 2.7. Extracted Measurements

Both pre- and post-operative processing pipelines output several clinically relevant metrics, some of which are depicted in Figure 6.

#### 2.7.1. Global Pre-Operative Metrics

Global metrics characterize the overall shape and size of the cochlea. These include the volume and surface area of the cochlea along with cochlear dimensions *A* and *B* originally proposed by Escude et al. [67], which are defined by the length of the straight line between the round window, passing through the modiolar axis, and reaching the furthest point around the $180^{\circ }$ cochlear angle and its perpendicular line, respectively (Figure 6b). Cochlear height *h* is computed along the modiolar axis. These measurements can be computed for the labyrinth or specifically for ST or SV. Cochlear shape is also defined by its potential “*rollercoaster*”, which represents the largest deviation in height from a linear fit of the spiral height—or the vertical “dip” of the basal turn before the cochlear spirals upwards around the modiolar axis [68]. Nautilus also supports automatic computations of cochlear, basal and two-turn duct lengths of the labyrinth, ST, and SV along various trajectories within these structures: along the estimated paths of the lateral wall (LW), modiolar wall (MW), organ of Corti (OC) and spiral ganglion (SG) [68] (Figure 6d). The extraction of these metrics allows the computation of the cochlear wrapping factor, which represents the logarithmic spiral angle of the cochlea, and the wrapping ratio, which represents the ratio of the maximum cochlear angle (at the helicotrema) and the lateral wall duct length.

#### 2.7.2. Local Pre- and Post-Operative Metrics

Local metrics characterize cochlear structures at particular places along the cochlear spiral. From pre-operative image processing, cochlear duct cross-sections are extracted at fixed angular displacements based on the labyrinth centerline. Cross-sectional area, radius, height, angle, minor and major axis lengths can then be computed by fitting an ellipse within each specific cross-section [69,70].

Post-operatively, registration parameters and the estimated locations of each electrode allow the computation of other important metrics. Electrode intracochlear positioning is characterized both by distance and angular measures at each electrode contact (where *RW* relates to $0^{\circ }$ and *Ap* corresponds to the maximum cochlear angle, which is typically around $900^{\circ }$) cochlear coverage). From these, the characteristic frequencies associated with each electrode are proposed in relation to OC [12] or SG [13,14]. In addition, the distance of each electrode contact to the MW and the estimated BM position are also computed.

### 2.8. Failure Flagging Mechanisms

Any automated system can occasionally fail. Transparency to the user (e.g., in the form of notifications or flags) in case of such failures is particularly important in order to identify which data point to exclude in any further observation or statistical analysis realized on Nautilus’ outputs. Therefore, Nautilus embeds a self-check flagging module that looks for signs of failures (e.g., detects suspicious segmentation or unexpected electrode array parameters) and explicitly notifies the user that images might not have been successfully processed and that the results should therefore be checked and/or used with caution. Whenever a flag is raised, a corresponding message is shown to the user (see Appendix A for an example). Specific flags have been implemented at each processing stage. They are presented in Table 1. Appendix A depicts the receiver operating curve (ROC) for the combined flags, based on which the cutoff values for notifying the user of a potentially faulty processing were chosen.

### 2.9. Data Export

The user can generate an export bundle containing all the outputs of the analysis in diverse export formats (Parquet, Excel, JSON) allowing further data analysis in their tool of choice. These analysis results are tagged with the unique version identifier for the specific processing pipeline version that was used for processing. Users may generate an export file for each case individually or a group of cases filtered on date. Appendix A presents distributions of cochlear metrics computed by our pipeline using the export.

## 3. Results

### 3.1. Evaluation Datasets

A well-curated multi-centric dataset, comprising both clinical and cadaver bones, was chosen for tye evaluation of Nautilus. CT images acquired from various scanners, using various acquisition parameters, and presenting heterogeneous resolutions, contrasts, and signal-to-noise ratios were included both for training and evaluation (see Appendix A). Groundtruth annotations, comprising the *C*, *Ap* and *RW* landmarks, cochlear structures and the electrode center points, were delineated by an expert radiologist using ITK-SNAP [64]. Limited by the poor resolution and imaging conditions of clinical images, only the cochlea could be manually delineated for clinical scans. On the other hand, ST and SV were successfully delineated in cadaver head CT scans since better contrast and resolutions could be achieved. The number of images used for training and evaluation for each process are mentioned in their respective sections. Each part of the pipeline was independently evaluated, as detailed below. A summary of the results is presented in Table 2.

### 3.2. Accuracy

#### 3.2.1. Landmark Detection

The landmark detection pipeline, utilized both pre- and post-operatively, was evaluated on a dataset of 60 images. The images were passed through the landmark detector, and the distance between the predicted and groundtruth annotation landmarks was computed. Mean detection errors of 0.71±1.0 mm, 0.75±1.14 mm, and 1.30±1.73 mm were observed for *C*, *Ap* and *RW*, respectively. All the individual errors were within a distance of two voxels, with the *RW* landmark yielding the worst performance.

#### 3.2.2. Segmentation

Nautilus’ segmentation pipeline was evaluated on four different clinical and cadaver datasets. The clinical dataset consisted of 58 pre-operative images with voxel resolutions ranging from 0.1 to 0.4 mm in the x-y plane and slice thickness ranging from 0.1 to 1 mm. The images were uploaded on Nautilus, and the union of ST and SV segmentation masks were obtained and compared with the manually labelled cochlea annotations. All the images were successfully processed, and a mean dice similarity coefficient and average surface error [71] of 86±3% and 0.14±0.03 mm were, respectively, observed for the clinical dataset. The cadaver datasets comprised 23 temporal bone (TB) µCT images in total. For computational limitations, the CT scans were resampled to an isotropic resolution of 0.1 mm. The images were uploaded on Nautilus, and the segmentation masks were obtained and compared with the manually labelled ST and SV annotations. All the images were successfully processed, and a mean dice similarity coefficient and average surface error of 80±3% and 0.19±0.04 mm were, respectively, observed for this cadaveric image dataset.

Figure 7 depicts segmentation results for each dataset. For a more thorough analysis, the cochlea was sectioned along its centerline at an $18^{\circ }$ angular interval. Dice similarity coefficients were computed for each segment (see Appendix A), where it appears that Dice scores decrease towards the apical area.

#### 3.2.3. Registration

The registration pipeline was evaluated on a dataset containing 15 sets of pre- and post-operative images with resolutions ranging from 0.1 to 0.3 mm. These image pairs did not necessarily have the same resolution. Each post-operative image of each pair was registered together with its pre-operative counterpart, and the average distances between the pre- and post-operative *RW*, *Ap*, and *C* landmarks within the registered coordinate system were computed to quantify the registration error. A mean target registration error of 0.88±0.39 mm was obtained.

#### 3.2.4. Electrode Detection

The electrode detection pipeline was evaluated on a dataset of 60 post-operative images. The electrode coordinates for each image were determined using Nautilus and compared with their corresponding groundtruth coordinates. An average electrode detection distance error of 0.09±0.16 mm was achieved for successfully processed images (those that did not were rated as failures as part of our failure detection analysis—see Section 2.8).

### 3.3. Robustness

A retrospective robustness analysis was carried out, in which two experts from the Hannover Medical School, Hannover, Germany, and the Institut de la Face et du Cou, Nice, France, independently verified the subjective quality of both pre- and post-operative analysis outputs. A dataset of 156 ears (81 left, 75 right) was used for this study. The reviewers were presented with an assessment sheet in which they reported their subjective evaluations of the quality of the input image (both pre- and post-operative), the quality of the segmentation, and the quality of the reconstruction of the electrode array. Reviewer 1 marked 87 pre- and 59 and post-operative images as being of “good quality”. The remaining pre-operative images were either classified as having poor resolution, being very noisy or already containing an electrode array. A total of 2 out of 156 cases were marked as failures, yielding a pre-operative processing success rate of 98.7%. For the post-operative assessment, 37 cases were marked as failures, yielding a success rate of 76.2%. However, a success rate of 88.3% was realized if out-of-specification images (images that the reviewers judged as being of poor quality) were excluded from the cohort. Reviewer 2 marked 126 pre- and 60 post-operative images as being of good quality. A total of 5 out of 156 cases were marked as failures, yielding a pre-operative success rate of 98.1%. For the post-operative assessment, 33 cases were marked as failures, yielding a success rate of 78.4% or 85.2% if images judged of poor quality by the reviewer herself were excluded from the cohort.

### 3.4. Failure Detection

The outputs of Nautilus’ flagging system were compared with the qualitative assessment of the two reviewers, as detailed in the previous section. Appendix A presents a performance summary of each flagging mechanism. An overall pre-operative failure detection sensitivity and specificity 100% and 97.4%, respectively, was achieved, with a corresponding post-operative failure detection sensitivity and specificity of 97.3% and 59.7%, respectively.

### 3.5. Computational Performances

Average computation times for each process are listed in Table 2. Computation times were obtained for a processing run on a standard Azure cloud VMs (Standard DS3 v2). On average, a complete pre- and post-operative analysis took around 10–12 min, with data storage and shape model adaptation for the segmentation taking the most time. All the other processes take less than two minutes combined. Nautilus is orchestrated with Azure Kubernetes with scalability in mind, and the throughput can be trivially scaled up by increasing the number of worker nodes.

## 4. Discussion

We present a web-based imaging research platform enabling the segmentation of cochlear structures and reconstruction of a cochlear implant electrode from conventional pre- and post-operative CT scans, respectively. Detailed analyses of accuracy, robustness, and failure detection provide legitimate grounds for using Nautilus for the exploration of clinically relevant questions on cochlear implantation and envisage further developments towards image-guided CI therapy.

Nautilus demonstrates segmentation performances in the range of previously presented academic results. More recent works have reported average cochlear Dice scores and average surface errors in the range of 72–91% and 0.11–0.27 mm, respectively [8,9,10,20,72]. Some of these groups have achieved higher Dice scores on limited datasets with high-resolution CT and µCT images [8,72]. A direct comparison between the works is not possible since our dataset and analysis focused on clinical and downsampled µCT images. Moreover, there is no publicly available benchmark analysis available for a fair comparison between different approaches. Nevertheless, our results on a varied dataset supports our claim of high accuracy and usability with conventional clinical CTs.

Many prior works have focused on inferring cochlea shape from µCT or high-resolution CTs as they offer good contrast and resolution compared to routine clinical CTs [8,72]. Our segmentation approach relies on JASMIN-inspired shape analysis [20], which offers the advantage of more interpretability of the estimated model parameters allowing further statistical studies. However, the same process is the bottleneck of our pipeline in terms of computational efficiency. This process could be adapted to benefit from learned shape models and anatomically inspired post-processing [73,74]. Our analysis also suggests that Nautilus performs better on clinical CT scans compared to cadaver head scans, which might be inherent to the cadaver head preparation process that often results in random air pockets, leading to a different intensity profile [75]. Additionally, our training dataset is comprised of mainly clinical scans. In future, a cadaver-specific pipeline may be developed to support cadaver-based research. Regardless, this is not a limiting factor in the applicability of Nautilus, as the main foreseen applications are in clinical research. Furthermore, our discretized analysis of the segmentation revealed that the performance decreases beyond two turns of the cochlea because of the small diameter of the cochlear ducts relative to image resolutions. This, however, is also not a limiting factor as most of the CI electrode arrays only reach around 450–600${}^{\circ }$ of insertion coverage.

Post-operatively, our electrode detection process outperforms previously reported works, which have reported localization errors in the range of 0.1–0.35 mm [58,61,62]. The electrode contact-BM distances could serve for inferring insertion trauma according to the Eshraghi trauma scales [55]. This would require distance-trauma evaluation against either cadaveric histology samples or high-resolution µCT scans where the various grades of BM trauma would be resolvable. We must note that metallic artifacts emanating from the electrodes do not permit direct segmentation of cochlear structures. This warrants the necessity of a pre-operative CT-scan to infer information about the cochlear structures. The post-operative images can be converted into pseudo-pre-operative images suitable for segmentation using artifact reduction techniques [76], or an atlas can be adapted on the post-operative to segment it directly [77]. The metallic artifacts might have an impact on pre-post registration as well. However, the challenge of post- to pre-operative image conversion can be circumvented by simply using a mirrored version of the contralateral cochlea in the post-operative scan if that contralateral ear is not implanted [24].

Although accuracy is an elementary performance metric for any segmentation pipeline, robustness is key for the usefulness of a tool such as Nautilus, especially given the heterogeneity of image quality expected to be input to the tool. Our subjective quality assessment provides an indication that Nautilus can be used with confidence when dealing with images of various resolutions, contrast, and signal-to-noise ratios. To the best of our knowledge, no other work in this domain has focused on robustness analysis from a comprehensive multi-centric dataset with varying image qualities. Recently, Fan et al. achieved 85% robustness for cochlea segmentation on their 177-image dataset [44]. Contrarily, our qualitative analysis depicts a robustness of around 97% with clinically reasonable performance. Our analysis enabled us to identify a resolution cutoff beyond which robustness seems to drop. The processing of images presenting voxel sizes superior to 0.3 mm does result in a significantly greater number of failures or inadequate outputs. This assessment, therefore, sets input specifications for recommended input image resolutions.

Because the probability of failure of our pipelines is non-zero, especially if out-of-specification images are input to the tool, Nautilus does provide cautionary flagging mechanisms that embody our guiding design principle of transparency. Our current set of flags has been 100 percent sensitive and about 60 percent specific, meaning that processing failures are very unlikely to go unaccounted for and that the system will result in false positives (notified non-failures) in less than half of the time, which we deemed an acceptable threshold for usability, especially as Nautilus is robust. A further observation for failures related to electrode detection in particular is that any failures are hard failures and easily noticed by the user. All in all, our flagging mechanisms should be useful to call for manual verification and potentially discard faulty analyses.

The set of features proposed by Nautilus provides legitimate grounds for exploring many relevant clinical and basic questions related to cochlear anatomy. Nautilus’ statistical model of the electrode insertion trajectory from pre-operative images, for instance, could be used prospectively to aim at a specific insertion angular coverage. The accuracy of these predictions could be validated using Nautilus with the post-operative images. Post-operatively, Nautilus makes possible the exploration of anatomo-physiologically-tuned fitting [78,79] or the exploration of the relationship between electrode geometrical configuration within the cochlea and clinical outcomes, including perhaps residual hearing. For all its utility, Nautilus could in the future be extended with additional features to address a broader spectrum of investigations, such as these related to the prediction of insertion difficulties during surgical planning, including for abnormal anatomies [80,81]. The delineation of other structures, including the facial nerve, chorda tympani, or RW would then be required. Other imaging modalities (e.g., MRI) and electrode arrays could be the subject of future developments. Bridging pre- and post-operative use-cases, an augmented reality setup inspired by [82] could be envisaged for intraoperative guidance.

## Figures and Tables

**Figure 1 jcm-11-06640-f001:**
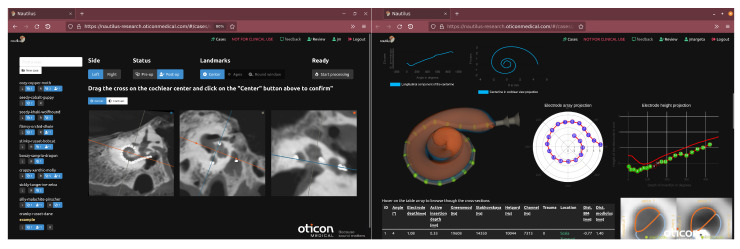
Nautilus offers a comprehensive set of research tools for pre- and post-operative cochlear image analysis for CI implantation and interactive visualization via a web browser. A number of metrics and additional outputs are generated by the pipeline and are made available for data export (e.g., spreadsheet of metrics for all cochleae in a user’s collection or STL models of the cochlear meshes) for further data analysis and applications (e.g., simulation or 3D printing, novel electrode array development).

**Figure 2 jcm-11-06640-f002:**
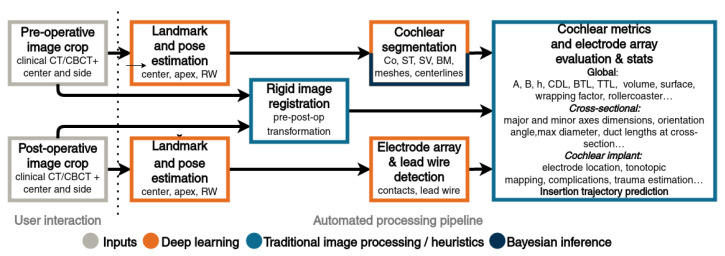
Nautilus pipeline overview. After the images are dropped onto a web browser window, the user moves a cross-hair roughly to the cochlea’s center and selects the side (left/right) and whether it is a pre- or a post-operative scan. A crop (10 × 10 × 10 mm) centered on that landmark is then rid of personally identifiable information and uploaded for processing. First, relevant landmarks (the center, round window, and apex) are estimated and used for initial cochlear pose (reference coordinate system) computation. Segmentation of the cochlear bony labyrinth (CO) is obtained through a convolutional neural network, whereas subsequently, the scala tympani (ST) and scala vestibuli (SV) are obtained using Bayesian inference. From the post-operative image, electrode array contact coordinates and lead wire are extracted and fit to the Oticon Medical EVO electrode CAD model. An interactive visualization as well as pre- and post-operative metrics are available directly on the web browser. A number of additional outputs are generated by the pipeline and made available for data export for further processing and applications. The segmentations can be exported in STL format for 3D printing, for instance. An estimate of electrode trajectory is also provided from the pre-operative image to estimate the equivalent angular coverage for a given electrode insertion depth in millimeters.

**Figure 3 jcm-11-06640-f003:**
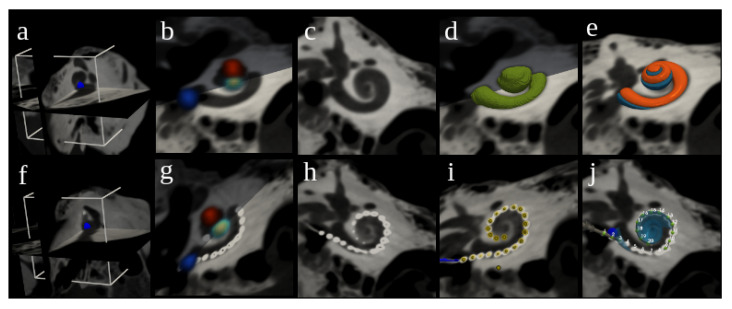
Steps of the image analysis pipeline in Nautilus. Regions of interest (10 × 10 × 10 mm) around a manually placed center (blue sphere) are cropped from both pre-operative (**a**) and post-operative (**f**) images. Landmark heatmaps are estimated (**b**,**g**) for the center (green), round window (blue), and apex (red). Images are aligned with rigid registration (**c**,**h**) as shown in cochlear view. Segmentation of the cochlear bony labyrinth (CO) (**d**) is subsequently split into the scala tympani (ST) and scala vestibuli (SV) (**e**). From the post-operative image, electrode array contact coordinates and lead wire are extracted (**i**), and an Oticon Medical EVO electrode CAD model is fit (**j**).

**Figure 4 jcm-11-06640-f004:**
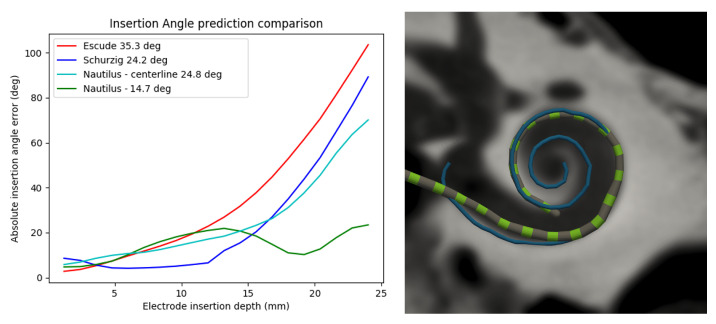
Angular insertion depth estimation based on the number of electrodes inserted inside the cochlea. The comparison graph shows (**left**), contrarily to our approach, how the performance of state-of-the-art approaches decreases exponentially with insertion depth. On the (**right**) is an example of a predicted trajectory (blue) and inserted electrode.

**Figure 5 jcm-11-06640-f005:**
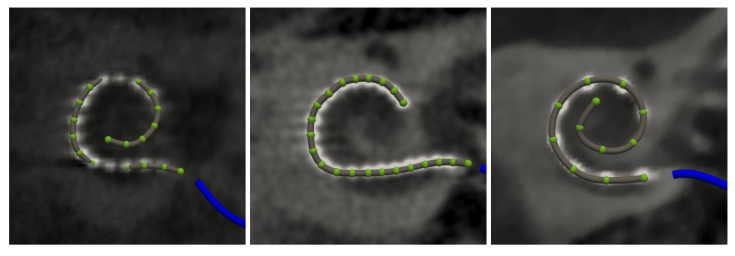
Electrode array detection in Nautilus has been developed and validated with Oticon Medical EVO electrode arrays (**left**) in mind. However, the same approach can be used with other electrode arrays. Example detection outputs for Cochlear Nucleus CI622 (**middle**), and MED-EL FLEX24 (**right**) cochlear implants.

**Figure 6 jcm-11-06640-f006:**
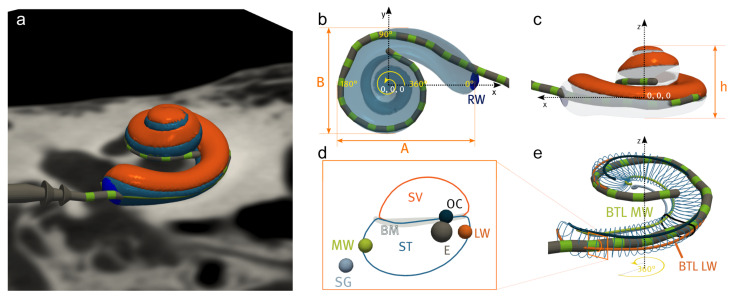
Cochlea and cochlear implant in the reference coordinate frame (**a**) and representation of different global (**b**,**c**) or cross-sectional metrics (**d**,**e**) that can be obtained using Nautilus. Examples include A, B, and the basal turn length (BTL) along various paths within the bony labyrinth (here, BTL LW and BTL MW are the 360-degree lengths covered while following the lateral wall (LW) or the modiolar wall (MW), respectively).

**Figure 7 jcm-11-06640-f007:**
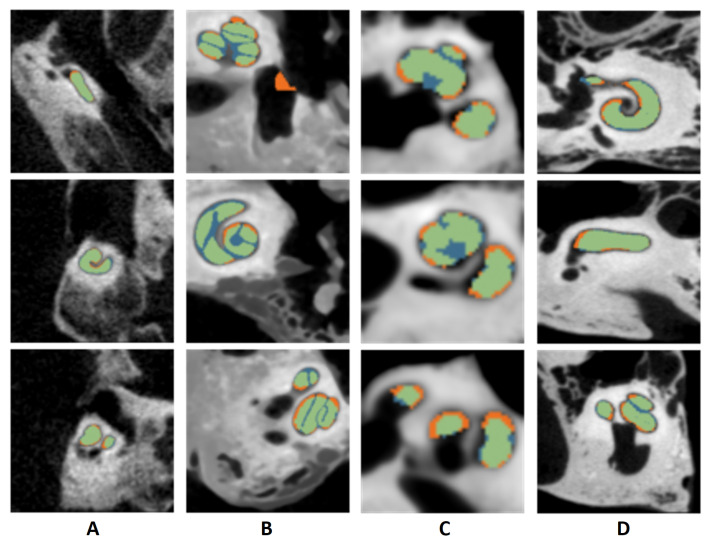
Segmentation output for different patients. (**A**) Clinical dataset, (**B**) cadaver dataset 1, (**C**) cadaver dataset 2, (**D**) cadaver dataset 3, blue: Nautilus estimation, orange: ground truth, green: overlap between the two.

**Table 1 jcm-11-06640-t001:** Description of different flags from the self-check module as implemented in Nautilus. The failure flags trigger a decreased level of confidence in the processed results when extra attention is needed.

Category	Flags Implemented
Image	poor image quality (resolution)
Segmentation	low cochlear volume
	low segmentation reliability
	irregular cochlear centerline
	irregular voxel intensities within segmented region
Registration	low correlation between pre-op and post-op
	large difference between registered landmarks
	too many electrode detected outside cochlea
	too many electrodes detected outside scala tympani
	non-basal electrodes detected outside cochlea
Electrode detection	incorrect number of electrodes detected
	irregular electrode ordering
	incorrect intensity at electrode locations
	irregular electrode pitch
	detected electrodes clustered together
	incorrect distance to modiolar axis
	electrodes detected near image boundaries

**Table 2 jcm-11-06640-t002:** Accuracy and robustness analysis for each pipeline process. ASSD: average symmetric surface distance, RAVD: relative absolute volume difference, HD95: 95% Hausdorff distance.

**Landmark Detection**												
**Dataset**	**Apex (mm)**				**Center (mm)**				**Round Window (mm)**			
Clinical (n = 60)	0.71				0.75				1.30			
**Segmentation**												
**Dataset**	**Dice (%)**			**ASSD (mm)**			**RAVD**			**HD95 (mm)**		
**Structure**	**CO**	**ST**	**SV**	**CO**	**ST**	**SV**	**CO**	**ST**	**SV**	**CO**	**ST**	**SV**
TB set 1 (n = 9)	83	67	64	0.17	0.21	0.18	−0.10	−0.02	−0.20	0.43	0.61	0.43
TB set 2 (n = 9)	77	64	58	0.21	0.23	0.24	−0.10	0.23	−0.38	0.76	0.77	0.99
TB set 3 (n = 5)	79	64	56	0.19	0.22	0.20	−0.21	−0.04	−0.40	0.62	0.71	0.64
Clinical (n = 58)	86			0.14			−0.13			0.35		
Mean	84	65	60	0.15	0.22	0.20	−0.14	0.02	−0.32	0.41	0.68	0.63
**Electrode Detection**												
**Dataset**						**Electrode Distance (mm)**						
Clinical (n = 60)						0.09						
**Registration**												
**Dataset**			**Mutual Information**					**Mean Registration Error (mm)**				
Clinical (n = 15)			0.15					0.88				
**Robustness Analysis**												
**Dataset**			**Reviewer 1 (%)**					**Reviewer 2 (%)**				
Pre-operative (n = 156)			98.7					98.1				
Post-operative (n = 156)			88.3 (76.2)					85.2 (78.4)				
**Failure Detection**												
**Dataset**		**Sensitivity (%)**			**Specificity (%)**				**Accuracy (%)**			
Pre-operative (n = 156)		100			97.4				97.4			
Post-operative (n = 156)		97.3			57.7				68.6			
**Computational Time**												
**Process**						**Approximate Time (s)**						
Landmark estimation						5.9						
Cochlear view generation						12.5						
Segmentation and pre-operative analysis						468.9						
Electrode detection and post-operative analysis						148.2						
Registration						49.8						

## Data Availability

The datasets analysed within the scope of the current study cannot be made publicly available as they have been made available to the authors under the specific authorization of the Hannover Medical School. The Hannover Medical School has collected the authorization of their patients to share their data anonymously for third-party analyses in the context of clinical research. This authorization does not extend to the public publication and distribution of the data. Access to the tool is, however, available upon reasonable request at nautilus_info@oticonmedical.com.

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
