# Peer review of "A Web-Based Automated Image Processing Research Platform for Cochlear Implantation-Related Studies"

_jcm, 2022, doi:10.3390/jcm11226640_

Round 1
Reviewer 1 Report
Nautilus is web-based research platform for automated pre- and post-implantation cochlear analysis which allows the automated, accurate, robust, and uncertainty-transparent delineation of the cochlea, scala tympani, scala vestibuli, and of the electrode arrays with tonotopic mapping from conventional Computed Tomography and Cone-Beam Computed Tomography images.
The importance to have the information about the cochlea metrics relevant to CI implantation such as the Cochlear Duct Length (CDL) which serves as an indicator of general cochlear size and depth of insertion are highly important. The CDL and other metrics also enable the computation of normalized tonotopic frequencies which will bring a great support for CI fitting, especially in bilateral implanted patients.
The proposed technology is long-awaited, will undoubtedly increase the effectiveness of cochlear implantation. The excellent work carried out by authors allows us to hope that in the near future the proposed approach can be widely used in clinical practice and will open new opportunities for scientific research.
Author Response
The proposed technology is long-awaited, will undoubtedly increase the effectiveness of cochlear implantation. The excellent work carried out by authors allows us to hope that in the near future the proposed approach can be widely used in clinical practice and will open new opportunities for scientific research.
Thank you for your review and the encouraging words.
Reviewer 2 Report
An interesting article that describes an excellent web-based device allowing the automatic segmentation of cochleae for research. The device shows robust efficacy in achieving its stated goals. Some questions remain on whether the BM determination used here is of sufficient resolution to be used for research, but that is not addressed in the current work.
My only note is that Figure 7 is missing the label for panel D.
Author Response
An interesting article that describes an excellent web-based device allowing the automatic segmentation of cochleae for research. The device shows robust efficacy in achieving its stated goals. Some questions remain on whether the BM determination used here is of sufficient resolution to be used for research, but that is not addressed in the current work.
Thank you for your review and your valuable feedback. We agree, the BM determination module will only benefit from further validation.
In this work, we constrain the BM by the previously validated shape model as mentioned in Section 1.4 (Segmentation of cochlear structures): "Instead of an active shape model, we build on top of a well validated Bayesian joint appearance and shape inference model [20,46]. The parameters of this shape model were tuned and validated on μCT data. The model can then serve as a strong prior constraining the final output for the lower resolution clinical CT images. This approach provides a probabilistic separation of ST and SV even in images of poor resolution. We provide an estimate of the BM location from the intersection of ST and SV's probability maps."
My only note is that Figure 7 is missing the label for panel D.
Thank you for pointing this out. We have added the missing label for panel D. The revised caption of Figure 7 on page 13 now reads as follows: “Segmentation output for different patients. A: Clinical dataset, B: Cadaver dataset 1, C: Cadaver dataset 2, D: Cadaver dataset 3, Blue: Nautilus estimation, Orange: Groundtruth, Green: overlap between the two”